# Identification and Characterization of the BZR Transcription Factor Genes Family in Potato (*Solanum tuberosum* L.) and Their Expression Profiles in Response to Abiotic Stresses

**DOI:** 10.3390/plants13030407

**Published:** 2024-01-30

**Authors:** Ruining Li, Bolin Zhang, Ting Li, Xuyang Yao, Tingting Feng, Hao Ai, Xianzhong Huang

**Affiliations:** Center for Crop Biotechnology, Anhui Science and Technology University, Chuzhou 239000, China; 18505525936@163.com (B.Z.); tinglizxy52174@163.com (T.L.); yaoxy@ahstu.edu.cn (X.Y.); fengtingtingyt@ahstu.edu.cn (T.F.); aihao@ahstu.edu.cn (H.A.)

**Keywords:** *Solanum tuberosum*, genome-wide of identification, *BZR*, abiotic stress, gene expression

## Abstract

Brassinazole resistant (BZR) genes act downstream of the brassinosteroid signaling pathway regulating plant growth and development and participating in plant stress responses. However, the *BZR* gene family has not systematically been characterized in potato. We identified eight *BZR* genes in *Solanum tuberosum*, which were distributed among seven chromosomes unequally and were classified into three subgroups. Potato and tomato *BZR* proteins were shown to be closely related with high levels of similarity. The *BZR* gene family members in each subgroup contained similar conserved motifs. *StBZR* genes exhibited tissue-specific expression patterns, suggesting their functional differentiation during evolution. *StBZR4*, *StBZR7*, and *StBZR8* were highly expressed under white light in microtubers. *StBZR1* showed a progressive up-regulation from 0 to 6 h and a progressive down-regulation from 6 to 24 h after drought and salt stress. *StBZR1*, *StBZR2*, *StBZR4*, *StBZR5*, *StBZR6*, *StBZR7* and *StBZR8* were significantly induced from 0 to 3 h under BR treatment. This implied *StBZR* genes are involved in phytohormone and stress response signaling pathways. Our results provide a theoretical basis for understanding the functional mechanisms of *BZR* genes in potato.

## 1. Introduction

A variety of endogenous and exogenous signals regulate plant growth and development. Plant hormones are major endogenous signaling molecules that can respond rapidly to environmental stimuli [1]. Brassinosteroids (BRs), a class of plant steroid hormones, play an important role in regulating plant growth and development, along with biotic and abiotic stress responses [2,3]. BRs fully stimulate the anti-stress potential of plants, alleviating damage caused when encountering various adverse external environmental cues through internal regulation [4]. Brassinazole resistant (BZR) is a family of transcription factors (TFs) that acts downstream of the BR signaling pathway, regulating plant growth and development and participating in plant stress responses by regulating the expression levels of some BR-responsive genes [5,6,7,8,9].

BZR proteins contain a nuclear localization sequence (NLS) at the N-terminal, a highly conserved DNA binding domain, a phosphorylation domain (which can be phosphorylated by BRASSINOSTEROID INSENSITIVE2-BIN2), a PEST sequence and a C-terminal domain [10]. BZR1 and Brassinosteroid insensitive 1-ethyl methanesulfonate-suppressor 1 (BES1) are two downstream TFs belonging to the BZR TF family [2]. BZR1 and BES1 share 88% amino acid sequence similarity, and the sequence consistency of the DNA binding domain is 97% [11]. They can be dephosphorylated by protein phosphatase when BR is detected [12]. Dephosphorylated BZR1 and BES1 accumulate in the nucleus and directly bind to *cis*-elements to regulate plant growth and development [13]. BZR1 and BES1 mediate crosstalk between BR and diverse signals such as other phytohormones, light, and stress, thereby regulating plant development and environment adaptability [5]. Both BZR1 and BES1 have a basic helix-loop-helix (bHLH) DNA binding motif in the N-terminal domain that is highly conserved across the whole family, although their functions have diverged [10]. BZR1 binds to a BR-Response Element (CGTGT/CG motif) to suppress the expression of BR-biosynthetic genes [14], while BES1 binds to an E box (CANNTG sequence) to activate BR induced gene expression [15].

The *BZR* gene family was first discovered and identified in *Arabidopsis thaliana* [10]. Research on BZR TFs in plant species has demonstrated their involvement in the regulation of cell elongation and division, plant morphology, flowering and fertility, quality improvement, and fruit ripening [16,17,18,19,20,21,22,23,24]. BZR1 directly regulates C-repeat binding factor 1 (CBF1) and CBF2 expression in *Arabidopsis*, thereby regulating freezing tolerance [25]. It was reported that AtBES1 plays a negative role in drought responses [26]. On the contrary, TaBZR2 positively regulates drought responses by activating TaGST1 [8]. BZR1/BES1 can directly bind to the promoters of several giberellin biosynthetic genes and control their expression in *Arabidopsis* and rice (*Oryza sativa* L.) [27,28]. Furthermore, TaBZR2 confers resistance to wheat stripe rust through the activation of chitinase Cht20.2 transcription [29]. SlBZR1 regulation of cold tolerance in tomato is related to its levels of phosphorylation [30]. BZR1 is a positive regulator of the BR signaling pathway and cooperates with light signal TFs to regulate cell elongation and plant growth [31,32]. *BZR* can also regulate the expression of drought-responsive glutathione s-transferase 1 (GST1) and interact with RESPONSIVE TO DESICCATION 26 (RD26) and WRKY TFs to modulate plant responses to drought, high temperature and freezing stress [33,34]. The *Solanaceae lycopersicum* BZR/BES TF SlBZR1 positively regulates BR signaling and salt stress tolerance in tomato and *Arabidopsis* [35]. In *Pyrus bretschneideri*, the *BZR* gene *PbBZR1* also acts as a transcriptional repressor of lignin biosynthetic genes in fruits [36]. Overexpression of *BpBZR1* enhances salt tolerance in *Betula platyphylla* [37]. *GhBZR3* suppresses cotton fiber elongation by inhibiting very-long-chain fatty acid biosynthesis [38]. TOPLESS (TPL) mediates brassinosteroid-induced transcriptional repression through interaction with BZR1 [39]. Nine BES1 genes localized on eight chromosomes were identified in potato [40]. Exogenous BRs application increased both the number and the total weight of potato tuber [41]. *StBRI1* is a functional potato BR receptor and has a novel function for brassinosteroid signaling in controlling tuberization [42]. The activation of reactive oxygen metabolism and phenylpropanoid metabolism by BR could accelerate the wound healing of potato tubers [43]. Transcription analysis during the sprouting process suggested that three BR signaling pathway genes, *delta 24 sterol reductase* (*DWF1*), *brassinosteroid c-6 oxidase* (*BRD1*) and *brassinosteroid-insensitive 1* (*BRI1*), are all upregulated in potato [33], sterol metabolism linked with *BZR* genes in potato has not been reported. BRs accelerate the conversion of starch into soluble sugar in tubers, contributing to sprouting in potato [44].

Potato is the fourth-largest major crop in the world and a critical component of the human diet in some countries [45]. The importance of potatoes in securing food and nutritional security has been identified by the Food and Agriculture Organization (FAO) of the United Nations. Potato genome sequencing has been completed. The development of potato genome provides facilitation for exploring the function of related genes in potato growth and development [46,47,48,49]. Previous studies have shown that BR is involved in regulating tuber sprouting [33]. However, studies on the function and regulatory mechanism of *BZR* in potato, involved in tuber development, are still limited. There is a lack of systematic genome-wide identification and functional analysis of potato BZR TF gene family members.

In this study, from the whole potato genome, bioinformatic methods were used to identify BZR TF gene family members, carried out chromosome mapping, analyzed systematic evolution, characterized structural characteristics of genes and proteins, investigated contraction and expansion of the gene family, and identified *cis* acting elements in promoter regions. In addition, the expression profiles of potato BZR TF genes in different tissues and in response to abiotic stresses and hormones were analyzed. Our findings will lay the foundation for the further functional analysis of BZR TF genes in tuber formation and development in potato, especially in response to abiotic stresses.

## 2. Results

### 2.1. Identification and Chromosome Distribution of BZR Gene Family Members

A total of eight *BZR* genes were identified in the potato genome following a genome-wide analysis conducted according to chromosomal location (Table 1, Figure 1). The eight *StBZR* proteins encoded contained one conserved domain (Appendix A). The amino acid sequence characteristics of *StBZR* family proteins revealed the *StBZR* proteins were composed of 315–695 amino acids. The StBZR6 amino acid sequence was the shortest at 315 aa, while that of StBZR1 was the longest at 695 aa. The proteins molecular weights ranged from 33, 905.1 to 78, 032.6 Da, with StBZR1 exhibiting the highest molecular weight (78,032.6 Da) and StBZR6 the lowest (33,905.1 Da). The theoretical isoelectric points ranged from 5.41 to 9.11. The prediction of protein localization results revealed that all *BZR* proteins are essentially localized in the nucleus (Table 1), which is consistent with the function of TFs.

Chromosomal mapping results showed the eight *StBZR* genes to be unevenly distributed on seven chromosomes (Figure 1). Chr01, chr03, chr04, chr07, chr08 and chr12 each harbored one *StBZR* gene, while two were located on chr02.

### 2.2. Phylogenetic Classification and Analysis of BZR Genes

To investigate the phylogenetic relationships between *BZR* family genes, the amino acid sequences of *BZR* proteins from *A. thaliana*, *S. tuberosum*, *S. lycopersicum*, *Nicotiana tabacum*, *O. sativa* and *Zea mays* databases were retrieved and phylogenetic trees using multiple sequence alignment were constructed (Appendix A). According to the topological structure of the phylogenetic tree, potato *BZR* proteins were divided into subgroups I, II and III (Figure 2). There were three *BZR* proteins in subgroup I, two *BZR* proteins in subgroup II and three *BZR* proteins in subgroup III. The *BZR* proteins from *S. tuberosum* and *S. lycopersicum* were found to be closely related, with high similarity.

### 2.3. Gene Structure and Protein Motifs of BZR Genes

Eight *BZR* proteins evolutionary tree revealed that these proteins cluster into three subgroups in potato (Figure 3A). A gene structure analysis showed that the *StBZR* genes contained 2–10 exons and 4–7 introns. Subgroups I, II and III contained the conserved BES1_N functional domain (Figure 3B). *StBZR1* and *StBZR7* in subgroup II were longer, and *StBZR2*, *StBZR5* and *StBZR8* in subgroup I were shorter. Although genes lengths differed in the same subgroup, the exon–intron lengths and gene structures were similar within each subgroup. The motif distributions in *BZR* proteins were analyzed, and 10 conserved motifs were predicted (Figure 3C,D). The high similarity in the motif structure of the *BZR* proteins in the same subgroup indicated that the *BZR* gene members is highly conserved. Six members contained motifs 1, 2 and 3, while *StBZR1* and *StBZR7* did not contain motif 3.

### 2.4. Contraction versus Expansion

To study the contraction and expansion of the *BZR* gene family during evolution, the collinearity of orthologous *BZR* genes from *A. thaliana*, *S. tuberosum*, *S. lycopersicum* and *N. tabacum* were analyzed. There were eight, eight, and six putative orthologous *StBZR* genes identified in *A. thaliana*, *S. lycopersicum*, and *N. tabacum*, respectively, while 12, 12, and eight pairs of collinear genes were identified in the comparison between *S. tuberosum* and *A. thaliana*, *S. lycopersicum*, and *N. tabacum*, respectively (Figure 4, Appendix A). These finding revealed that *StBZR* family genes number has expanded. The paralogous *BZR* gene pairs analysis showed that *StBZR* has two paralogous gene pairs in the potato genome, namely *StBZR5*/*8* and *StBZR3*/*4*, with each gene located on a different chromosome (Appendix A).

### 2.5. Expression Patterns of StBZR Genes in Different Tissues by qRT-PCR

The expression characteristics of eight *StBZR* genes in seven tissues were analyzed. The expression levels of eight *StBZR* genes were significantly different among the various tissues (Figure 5). *StBZR3* and *StBZR5* expression patterns were similar, with high expression levels in stem and low expression in other tissues. Moreover, the expression patterns of *StBZR2*, *StBZR6*, and *StBZR8* were similar, with the highest expression levels observed in petiole, followed by stem and tuber. The expression patterns of *StBZR4* and *StBZR7* were similar, with the highest expression levels seen in petiole and tuber. The expression of *StBZR1* in tuber was higher than that in other tissues. In summary, *StBZR* genes were highly expressed in petiole, stem, and tuber, while *StBZR1* was significantly specifically expressed only in tuber.

### 2.6. Distribution of Cis-Reactive Elements on the Promoter of StBZR Genes

To investigate the cis-acting elements in the promoter regions of *StBZR* genes, approximately 2000 bp of sequence upstream of the translation initiation site were analyzed. The analysis revealed several regulatory elements related to the induction of phytohormones such as ABRE involved in abscisic acid (ABA), TGA element involved in auxin and the P-box and GARE-motif involved in gibberellin (GA) responsive elements. The *cis*-acting elements also included Sp1, G-box, and Box 4 involved in light responsiveness, the CGTCA-motif involved in MeJA-responsiveness, CAT-box related to meristem expression, ARE element essential to anaerobic induction, and the TCA-element involved in salicylic acid responsiveness (Figure 6A). In addition, the *cis*-acting elements related to drought and salt responses (Figure 6B). The results showed that all of the *StBZR* gene promoter regions contained various *cis*-regulatory elements involved in light, stress and hormone-responsiveness, which implied that the *StBZR* family may participate in many growth and development processes. Therefore, we analyzed the gene expression patterns under light, drought, salt, ABA, and BR treatments to check the *StBZR* genes in response to abiotic stresses and hormone.

### 2.7. Expression Profiles of StBZR Genes in Response to Abiotic Stresses and Hormones by qRT-PCR

We examined the expression patterns of the *StBZR* genes under abiotic stresses in the presence of phytohormones. Plants were treated with various wavelengths of light, drought, salt, ABA, and BR. The leaf development of the potato plantlets was severely inhibited under monochromatic red light, there was no leaf sample under red light. The expression levels of *StBZR1*, *StBZR3*, *StBZR4*, *StBZR7*, and *StBZR8* under blue light were higher than under white light in leaf, and in microtubers, the expression levels of *StBZR4*, *StBZR7*, and *StBZR8* under white light were higher than under red and blue light (Appendix A). Under drought (15% PEG6000) treatment, *StBZR1* showed progressive up-regulation from 0 to 6 h and progressive down-regulation from 6 to 24 h. *StBZR8* was down-regulated under drought treatment (Figure 7). *StBZR1*, *StBZR2*, *StBZR5*, *StBZR6*, *StBZR7*, and *StBZR8* were first up-regulated and then down-regulated under salt treatment (150 mM NaCl) (Figure 8). Under treatment with 100 μM ABA, *StBZR4* was strongly induced, *StBZR3* was strongly and rapidly inhibited from an early stage, and then induced after 12 h (Figure 9). Under 50 μM BR treatment, *StBZR1*, *StBZR2*, *StBZR4*, *StBZR5*, *StBZR6*, *StBZR7,* and *StBZR8* were significantly induced from 0 to 3 h and then inhibited. In particular, the expression of *StBZR2*, *StBZR5*, and *StBZR8* was increased about 10-fold after 3 h. *StBZR3* was first down-regulated and then up-regulated (Appendix A).

## 3. Discussion

The *BZR* gene family is an important TF family that regulates plant growth, development, and the BZR-mediated abiotic stress response. However, the BZR TF family members in *S. tuberosum*, an important and widespread food and vegetable crop, have not been thoroughly investigated to date. In this study, we characterized the *BZR* genes in *S. tuberosum* and discovered discrepancies and variations in gene sequences, structures, and conserved motifs. We also identified instances of both the conservation and divergence of gene and protein expression in this family. Expression data analysis and *cis*-regulation prediction further revealed that potato *BZR* genes may be intrinsically involved in regulating pathways associated with plant development and stress resistance. The genome-wide identification and characterization of BZR TF family members in *S. tuberosum* is an essential starting point for in-depth exploration of the function of this gene family. The accumulation of genomic and transcriptomic data from *S. tuberosum* will provide insights into the functional and molecular characteristics of the *BZR* gene family.

BZR signaling components are highly conserved between primitive to modern plants [14]. Although plant genomes differ in size and numbers of genes, many *BZR* gene family members have been identified. For instance, six *BZR* genes have been characterized in *A. thaliana* [10], 11 in *Z. mays* [50], six in *Beta vulgaris* [51], nine in *S. lycopersicum* [52], five and two in wheat and foxtail millet, respectively [5], and six in *Cucumis sativus* [11]. In our study, a total of eight *BZR* genes were identified in *S. tuberosum* (Table 1), and these were unevenly distributed on seven chromosomes (Figure 1). The number of gene family members has shrunk or expanded in different species [5,10,11,50,51,52], due to environmental adaptations that have occurred during evolution.

Plant regulatory genes evolve quickly, but the rates at which different domains of *BZR* family proteins have evolved differ substantially. Subgroup I, II, and III members have no similarities in gene structure, and may have evolved from different origins. In our study, it was found that the *StBZR* gene does not belong to the intron-enriched gene family, so it is inferred that the continuous diversification of the *StBZR* gene family may lead to intron loss [53]. Group I of the *BZR* genes in potato contains three *StBZR* genes, group II contains two *StBZR* genes, and group III contains three *StBZR* genes. The gene structure in groups I and III is relatively conserved, while that in group II is considerably different (Figure 3), indicating that expansion of the *BZR* gene family has mainly occurred in subgroup III. To investigate the evolutionary relationship between *BZR* family members among species, we included *BZR* proteins from *S. tuberosum*, *S. lycopersicum*, *N. tabacum*, *O. sativa*, and *Z. mays* in a phylogenetic analysis and identified the conserved motifs. Cucumber and tomato were found to be closely related [11], while the phylogenetic trees of potato and tomato (*S. lycopersicum*) were shown to be similar (Figure 2). This result is consistent with the taxonomic relationships between potato and tomato within the *Solanaceae*.

BZR family genes are specifically expressed in different tissues and organs of many plants [34]. In *Arabidopsis*, the transcriptional expression of *BZR* genes is higher in roots and buds but lower in stems, fruits, and flowers [15,54], while in maize, these genes are highly expressed in seedlings and endosperm [50]. In tomato, *SlBZR2* and *SlBZR9* are generally expressed at high levels in most tissues, while other *SlBZR* genes are highly expressed only in certain tissues [55]. In our study, there were significant differences in the expression levels of the eight *StBZR* genes among various tissues in potato (Figure 5). The expression patterns of *StBZR3* and *StBZR5* were similar, with high and low expression levels seen in the stem and other tissues, respectively. *StBZR2*, *StBZR6,* and *StBZR8* were highly expressed in the petiole, followed by the stem and tuber (Figure 5). The differential expression of *StBZR* family members in different tissues indicates that these genes play a regulatory role in the growth and development of potato, and that there is likely to be functional redundancy. BRs are not only involved in starch utilization but also in the regulation of sucrose transport [56]. In sugar beet, *BZR* genes are involved in accumulating sugars in the taproot, eventually leading to increases in root diameter and weight [51]. The expression of *StBZR1* was demonstrated to be higher in the tuber than other tissues (Figure 5), indicating that this gene is could be involved in promoting sugar accumulation and thereby tuber expansion.

The differential pattern of *StBZR* gene expression in various tissues (Figure 5) suggested that these genes may act as growth regulators. An analysis of the promoter regions of the *StBZR* genes identified in this study revealed the existence of a variety of *cis*-acting elements that are involved in regulating the temporal and spatial expression levels of the genes (Figure 6). The *StBZR* gene promoter regions contained various *cis*-regulatory elements involved in light, stress and hormone-responsiveness, which implied that the *StBZR* family may participate in growth and development processes by responding to different signals (Figure 6). Differential expression patterns among the *StBZRs* were also observed following light spectrum, drought, salt, ABA and BR treatments. Thus, *StBZR* genes may participate in the regulation of plant development and abiotic stress resistance pathways. Some reports have confirmed that the *BZR* family is involved in a variety of hormone signaling pathways [57,58,59], stress responses [7,25,34], and plant growth regulation [60,61]. The analysis of gene expression suggested that some *StBZR* members participate in multiple stress responses. For example, the expression levels of *StBZR4*, *StBZR7*, and *StBZR8* under white light were higher than those under red and blue light, in microtubers (Appendix A). Under drought and salt stress, *StBZR1* was up-regulated first and then down-regulated, showing progressive up-regulation from 0 to 6 h and progressive down-regulation from 6 to 24 h after treatment (Figure 7 and Figure 8). *StBZR4* was strongly induced under ABA treatment (Figure 9), whereas, under BR treatment, *StBZR1*, *StBZR2*, *StBZR4*, *StBZR5*, *StBZR6*, *StBZR7*, and *StBZR8* were significantly induced from 0 to 3 h and then inhibited (Appendix A). This implied that these *StBZR* genes are involved in phytohormone and stress response signaling pathways. These data provide valuable insights into the functional mechanisms of the *StBZR* gene family in response to phytohormones and different biotic and abiotic stresses during plant development.

## 4. Materials and Methods

### 4.1. Plant Materials

Plantlets of potato (cv. *Shepody*) were provided by Anhui Science and Technology University. Plantlets of potato were cut into 1–1.5 cm segments with leaves. Then placed them into medium in tissue culture bottles (63 mm of inner diameter; 85 mm of height). Four stem segments were placed in each tissue culture bottle and were placed in a tissue culture room. The medium used for propagation of plantlets in vitro was solid Murashige and Skoog (MS) medium (4%, *w*/*v*, sucrose, 0.9%, *w*/*v*, agar). The medium used for the induction of microtubers was solid MS medium (8%, *w*/*v*, sucrose, 0.9%, *w*/*v*, agar). The relative humidity was 65 ± 5%, day-time temperature was 22 ± 2 °C, night temperature was 18 ± 2 °C, photoperiod uniformly was set to 8-h light/16-h dark and photosynthetic photon flux density was 65 μmol m^–2^ s^–1^ in tissue culture room. Potato plantlets grown for 30 d were acclimated and transplanted to pots containing vegetative soil and vermiculite (1:1) and placed in a plant growth room where the relative humidity was 65 ± 5%, the temperature was 22 ± 2 °C, the light intensity was 200 μmol m^–2^ s^–1^ and the photoperiod was set at 12-h light/12-h dark. 

### 4.2. Identification of BZR Transcription Factor Gene Family Members in S. tuberosum

We downloaded the genome sequence, proteins and corresponding coding sequences of *S. tuberosum* (Version 6.1) from the Phytozome v13 website (https://phytozome.jgi.doe.gov/pz/portal.html, accessed on 1 September 2023). The BES1_N domain file (PF05687) from the InterPro website (www.ebi.ac.uk/interpro/entry/pfam/PF05687/, accessed on 1 September 2023) were downloaded and uploaded it to the HMMERv3.3.2 (https://www.ebi.ac.uk/Tools/hmmer/, accessed on 1 September 2023) to search for potential genes in the potato genome containing this conserved domain, with an *E* < 1 × 10^−5^ [62]. The sequences of *A. thaliana* and *O. sativa*
*BZR* family members were downloaded from uniport (https://www.uniprot.org/, accessed on 1 September 2023) and compared with potato protein sequences using blast-2.11.0, with an *E* < 1 × 10^−5^. Based on the above method, putative *BZR* candidates were selected. After removing redundant results, the remaining sequences were further verified for the existence of BES1_N domains using other databases: Simple Modular 132 Architecture Research Tool (SMART, http://smart.emblheidelberg.de/, accessed on 1 September 2023), NCBI Batch CD-Search Tool (https://www.ncbi.nlm.nih.gov/Structure/bwrpsb/bwrpsb.cgi, accessed on 1 September 2023) and the sequences with lacking BES1_N domains were removed. Finally, the protein sequences with BES1_N domains were taken and named sequentially according to their locations on the chromosomes. We used the website Cell—Ploc 2.0 (http://www.csbo.sjtu.edu.cn/bioinf/Cell-PLoc-2, accessed on 1 September 2023) for predicting the subcellular localization of candidate genes encoding *BZR* proteins [63]. We used the online software ExPASy (Version 3.0, https://web.expasy.org/protparam/, accessed on 1 September 2023) to analyze the physicochemical properties of the amino acids encoded by the candidate genes.

### 4.3. Chromosomal Location

We used TBtools v1.09876 software [64] to calculate the location of each *StBZR* gene and length information of chromosome. We used MG2C (http://mg2c.iask.in/mg2c_v2.0/, accessed on 6 September 2023) to construct the physical location of *StBZR* genes on the chromosomes.

### 4.4. Phylogenetic Tree Reconstruction, Gene Structure and Protein Motifs Analysis

Multiple sequence alignments were performed on the potato *BZR* protein sequences using ClustalW [65] with default parameters. *BZR* protein sequences of *O. sativa*, *A. thaliana*, *Z. mays*, *N. tabacum*, and *S. lycopersicum* were downloaded from NCBI (https://www.ncbi.nlm.nih.gov/, accessed on 3 September 2023), and the *BZR* protein sequences of potato and these species were used to reconstruct a phylogenetic tree. We used MEGAX 10.1.8 software [66] and the neighbor-joining method, with the bootstrap value set to 1000 cycles to reconstruct the phylogenetic tree of the *BZR* protein family members. The *BZR* member gff3 file was submitted to GSDS (http:/gsds.cbi.pku.edu.cn, accessed on 3 September 2023) for gene structure analysis [67]. We used the MEME (http://meme-suite.org/, accessed on 4 September 2023) website for the motif prediction, with the number set to 10. Batch CD-search (https://www.ncbi.nlm.nih.gov/Structure/bwrpsb/bwrpsb.cgi, accessed on 6 September 2023) was used to analyze conserved domain structure.

### 4.5. Analyses of Duplication Type and Synteny

We used Blast-2.13.0 + for comparison, and MCScanX [68] to calculate collinearity of homologous *BZR* genes within and between species, with a threshold of *E* < 10^−5^. Using MEGAX [66] analyzed multiple sequence alignment of *StBZR* members. Using TBtools v1.09876 software [64] showed the collinearity of *BZR* among *A. thaliana*, *O. sativa*, *N. tabacum*, *S. lycopersicum* and *S. tuberosum* to judge collinearity of the *BZR* gene families.

### 4.6. Tissue Expression Characteristics of the StBZR Genes in S. tuberosum

The leaves, petioles, stem sections grown for 30 dflower tissues grown for 45 days and stolons, roots, tubers grown for 60 d were collected under plant growth room, performing three biological replicates. These tissues were immediately flash-frozen in liquid nitrogen and stored at −80 °C for RNA extraction and gene expression analysis.

### 4.7. Cis-Regulatory Element Analysis of the StBZR Genes

We extracted sequences in the 2000 base pairs (bp) upstream of the start codon (ATG) in the *BZR* genes from the genome of *S. tuberosum*. For promoter cis-acting regulatory element screening, we analyzed the sequences extracted above using the PlantCARE (http://bioinformatics.psb.ugent.be/webtools/plantcare/html/, accessed on 15 September 2023). Using TBtools v1.09876 software [64] drew the *cis*-elements in the promoter region.

### 4.8. Light Spectrum, Abiotic Stress and Hormone Treatments

Robust and uniform potato seedlings, which had been grown in test tubes for 30 d were selected for this study. The stems were cut into 1–1.5 cm segments and each section was placed in a separate glass test tube under aseptic conditions. The tubes were cultivated under dark conditions for 2 days and then grown under blue light at a wavelength of 460 nm, red light at a wavelength of 620 nm, and white light. Leaves and stolons of potato plantlets grown for 35 d and microtubers grown for 80 d were collected. Drought and salt were set as abiotic stress treatments. All reagents are purchased from Shanghai MACKLIN Co., Ltd, China. PEG 6000 and NaCl were added to 1 L of nutrient solution to prepare 15% PEG 6000 and 150 mM NaCl solutions, respectively. ABA and BR were dissolved in 100 mL distilled water to produce 1 mM solutions, which were then diluted accordingly. Hormone treatments involved the application of ABA (100 μM) and BR (50 μM). We collected samples at 0, 3, 6, 12, and 24 h post-treatment, and untreated seedlings from the same batch were used as controls. Three biological replicates were collected at each time point, with each replicate consisting of three independent seedlings.

### 4.9. qRT-PCR

We performed RNA extraction and reverse transcription as described by Huang et al. (2017) [69]. We performed qRT-PCR according to Jin et al. [70]. Using an ABIViiA7 real-time PCR instrument (Life Technologies, Carlsbad, CA, USA) performed qRT-PCR. Using Primer 3.0 tool (https://bioinfo.ut.ee/primer30.4.0/, accessed on 10 September 2023) designed the *StBZR* gene-specific amplification primers. We used *elongation factor*-*1alpha* (*EF1α*) as a reference gene [71] (Appendix A). We performed three independent biological replicates of qPCR, and each PCR reaction was performed in triplicate. The 2^–∆CT^ method was used to calculate the relative expression levels of genes in different tissues. The 2^–∆∆CT^ method was used to calculate the relative expression levels of genes under light spectrum, abiotic stress and hormone treatments [72]. SPSS software (Version 20) was used for the statistical analysis. Significant differences among the groups were compared based on Tukey’s test (*p* < 0.05). GraphPad Prism 9.5.0 software was used for drawing of data.

## 5. Conclusions

Eight *BZR* genes were identified from the genome of potato. These classify into three subgroups and are distributed on seven chromosomes unevenly. *StBZR* and SlBZR proteins are closely related phylogenetically and display high sequence similarity. 12 genes were orthologous to the *BZR* genes in *S. tuberosum* and *S. lycopersicum*. Tissue specific expression characteristics suggest functional differentiation of *StBZR* genes during evolution. *StBZR* gene promoters contain many regulatory elements that are involved in phytohormone, light and stress signaling. This study offers a basis to predict the functions of *BZR* genes in potato, and lay a foundation for further research of the biological functions of *BZR* genes in potato. In order to further deepen our understanding of the function of *BZR* genes, our future direction will focus on the study of the specific regulatory mechanisms of *BZR* genes in regulating potato tuber formation and development.

## Figures and Tables

**Figure 1 plants-13-00407-f001:**
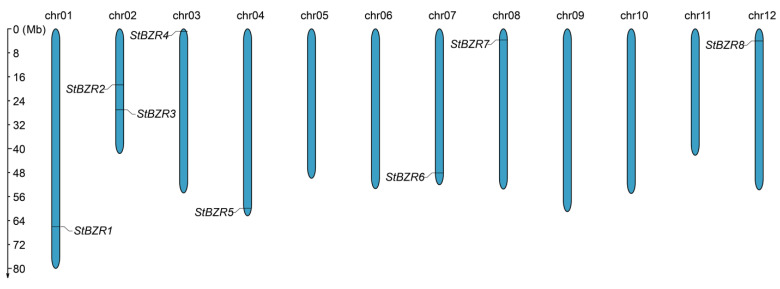
Distribution of the *StBZR* genes on chromosomes. The left scale indicates chromosome length.

**Figure 2 plants-13-00407-f002:**
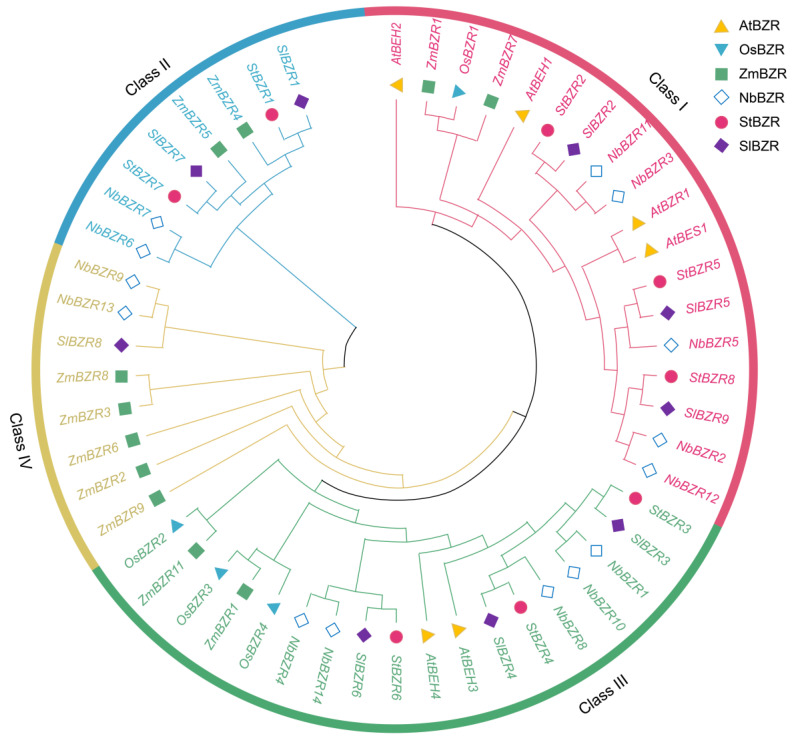
A phylogenetic tree of *BZR* family genes from *S. tuberosum* (St), *A. thaliana* (At), *S. lycopersicum* (Sl), *N. tabacum* (Nb), *O. sativa* (Os) and *Z. mays* (Zm).

**Figure 3 plants-13-00407-f003:**
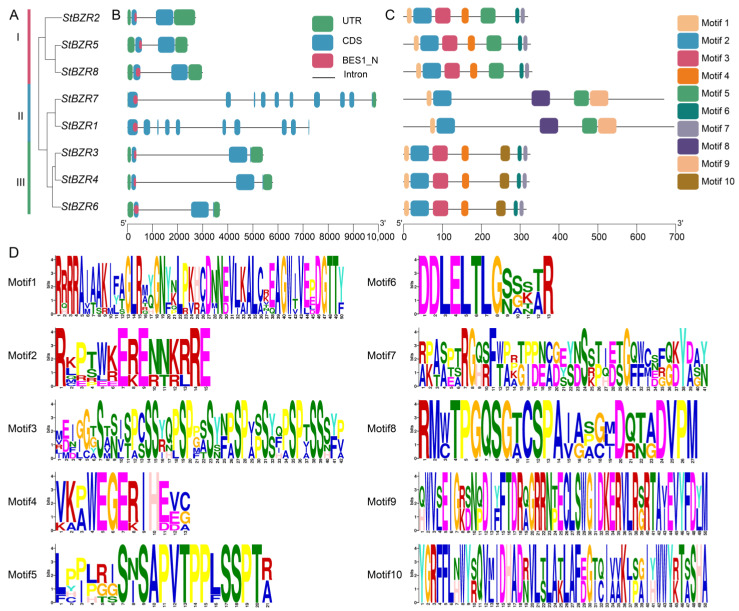
The distributions of motif and exon-intron structures for *BZR* family members in *S. tuberosum*. (**A**) phylogenetic tree constructed using the *StBZR* protein sequences; (**B**) gene structure of *StBZR* members; UTR, untranslated region, represented by green boxes; CDS, coding sequence, represented by blue boxes. (**C**) ten types of conserved motifs were predicted in the *StBZR* protein sequences. (**C**) represent protein length with specific regions of conserved domains. (**D**) The sequence logo conserved motif of the potato *BZR* proteins.

**Figure 4 plants-13-00407-f004:**
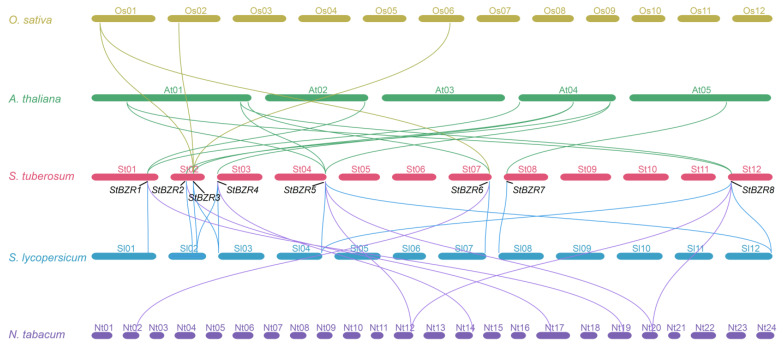
Syntenic relationships analysis between *BZR* genes in *O. sativa*, *A. thaliana*, *S. tuberosum*, *S. lycopersicum* and *N. tabacum*. Yellow, green, blue and purple lines highlight syntenic *BZR* gene pairs.

**Figure 5 plants-13-00407-f005:**
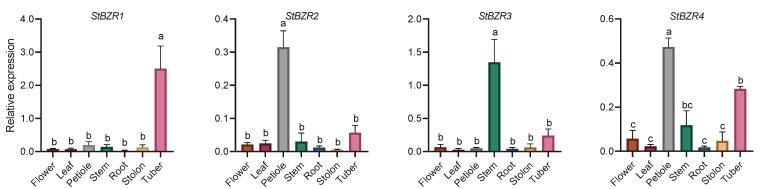
Expression patterns of the eight *StBZR* genes in seven different tissues analyzed using qRT-PCR. Different letters indicate significant differences between different tissues. Significant differences among the groups were compared based on Tukey’s test (*p* < 0.05). The data points represent mean ± SD.

**Figure 6 plants-13-00407-f006:**
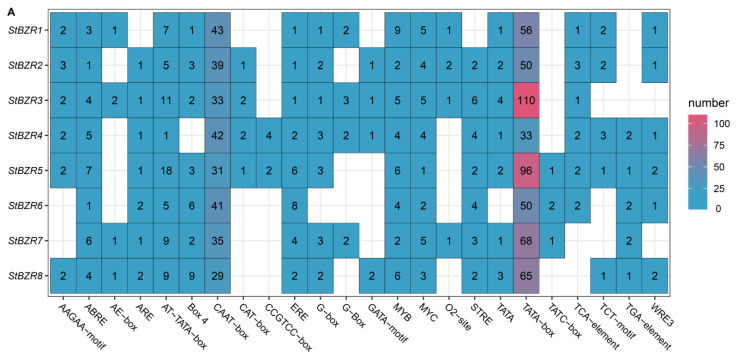
Distribution of *cis*-elements in the promoter regions of *StBZR* genes. (**A**) Heat map representing *cis*-acting components. Colors represent different types of elements; and (**B**) distribution of *cis*-reactive elements in the promoter. The ruler at the bottom indicates the direction and length of the sequence.

**Figure 7 plants-13-00407-f007:**
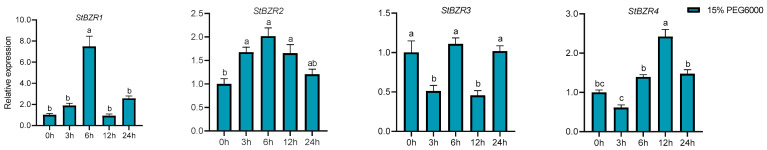
Expression patterns of the eight *StBZR* genes under drought (15% PEG6000) stress. Different letters indicate significant differences between different tissues. Significant differences among the groups were compared based on Tukey’s test (*p* < 0.05). The data points represent mean ± SD.

**Figure 8 plants-13-00407-f008:**
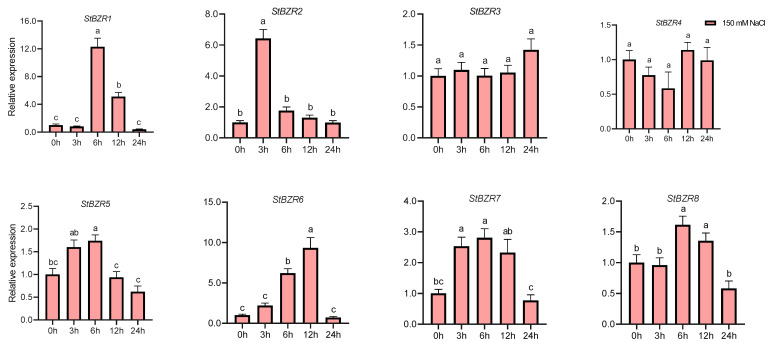
Expression patterns of the eight *StBZR* genes under salt (150 mM, NaCl) stress. Different letters indicate significant differences between different tissues. Significant differences among the groups were compared based on Tukey’s test (*p* < 0.05). The data points represent mean ± SD.

**Figure 9 plants-13-00407-f009:**
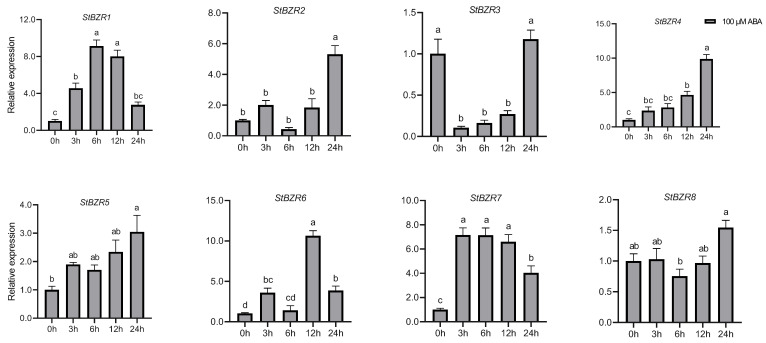
Expression patterns of the eight *StBZR* genes after 100 μM ABA treatment. Different letters indicate significant differences between different tissues. Significant differences among the groups were compared based on Tukey’s test (*p* < 0.05). The data points represent mean ± SD.

**Table 1 plants-13-00407-t001:** Profiles of the *BZR* gene family members identified in *S. tuberosum*.

Gene	Gene ID	Predicted Amino Acid Number/aa	Molecular Weight/Da	Theoretical Isoelectric Point	Predicted Localization
*StBZR1*	Soltu.DM.01G033560	695	78, 032.6	5.41	Nucleus
*StBZR2*	Soltu.DM.02G006820	319	34, 481.9	9.08	Nucleus
*StBZR3*	Soltu.DM.02G015130	325	35, 191.0	8.08	Nucleus
*StBZR4*	Soltu.DM.03G001120	323	34, 696.7	8.18	Nucleus
*StBZR5*	Soltu.DM.04G034930	326	34, 962.1	8.88	Nucleus
*StBZR6*	Soltu.DM.07G023410	315	33, 905.1	9.11	Nucleus
*StBZR7*	Soltu.DM.08G003130	669	75, 575.7	5.95	Nucleus
*StBZR8*	Soltu.DM.12G005470	330	35, 566.7	8.86	Nucleus

## Data Availability

All data analyzed during this study are included in the Appendix A.

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
