# Peer review of "Identification and Characterization of the BZR Transcription Factor Genes Family in Potato (Solanum tuberosum L.) and Their Expression Profiles in Response to Abiotic Stresses"

_plants, 2024, doi:10.3390/plants13030407_

Round 1
Reviewer 1 Report
Comments and Suggestions for Authors
Suggested corrections.
Figures 5, 7 and 8. In order to improve the legibility, the label of the x-axis sould be put also on all histograms of upper part of the picture not only at the bottom.
Tables 1 and S1. Considering that the number of amino acids for each gene is only in silico predicted, the label "Predicted number of amino acids" instead of "Amino Acid Number/aa" would be more precise.
Figure S3. Typo in the wording: "Bule" instead of "Blue". Moreover, it is not clear why for the red light there is not any data on the leaf for any of the described gene member of the family. If it not a mistake in the picture, and its just not expressed it must be written in the text
Figures 7, 8, S4 and S5. In order to improve the legibility, the label of the x-axis (timing of the time-course) sould be put also on all histograms of upper part of the picture not only at the bottom.
Table S2. The table should be improved by adding more information about the primers such as: %GC, predicted Tm, specify the forward and the reverse primer.
Text of the paper.
line 88. "bioinformatic methods" instead of "bioinformatics methods"
line 149. because all of the data presented were in silico predicted the expression "putative orthologous" would be much more appropriate.
line 325. Typo in first word: "uniport" instead of "uniprot".
line 398. Suggestion to add: the "specific" amplification of the StBZR genes.
lines 275-276. Considering that all the data presented are in silico predicted, and also the presupposition in the previous lines for other species are for an "involment" of putative omologous genes, a more prudent sentence would more appropriate: "(Figure 5), suggesting that this gene could be involved in tuber development of potato".
Reviewer 2 Report
Comments and Suggestions for Authors
In Fig5, gene expression is depicted across different tissues. It is not specified in the figure which tissue represents the standard for StBZR gene expression. Furthermore, it's important to note that various tuber tissues may be sampled depending on the maturity of the tuber, and gene expression can vary accordingly. It's advisable to check the supporting details or methods section to understand how the data accounts for these variations.
In Fig.7, the title of the paper implies that the StBZR family responds to abiotic stress. While the PEG treatment is presented, it raises the question of whether it is sufficiently representative. To establish a comprehensive understanding, it is recommended to explore various abiotic stresses through promoter analysis. This would provide a more thorough assessment of the StBZR family's response under diverse stress conditions.
In Figure 8, the authors aimed to demonstrate the response of the StBZR gene family to abiotic stress and ultimately determine gene expression through brassinosteroid (BR) treatment. Although different gene expression patterns can be informative, the experiment presented in Figure 8 seems somewhat unrelated to the main topic. These unrelated experiments should be moved to Supporting Information.
It would be better if Figure S4 and Figure S5 in the supporting data were used as main data along with PEG processing experiments.
Overall, the authors appear to lack a cohesive experimental design to effectively demonstrate the involvement of StBZR family genes in abiotic stress. In Figure 6, where various cis-elements are presented, there is an opportunity to better connect this information with the narrative related to abiotic stress. However, it seems that the abiotic stress experiments presented are somewhat disjointed and unrelated to the information gathered about cis-elements. Strengthening the connection between these elements would enhance the overall coherence and effectiveness of the study.
What is the meaning of TATA-box, the Cis-element that appears most frequently in Figure 6A?
At a minimum, if the authors identified the StBZR family, they should also include actual experiments for the predicted subcellular localization obtained in Table 1 in their results.
The tuber is the most important part of potatoes. Therefore, experiments on how each BZR gene changes are most important, but the authors show this only at simple time points.
Reviewer 3 Report
Comments and Suggestions for Authors
I have reviewed the manuscript intended to be published in Plants mdpi journal entitled:
Identification and Characterization of the BZR Transcription Factor Genes Family in potato (Solanum tuberosum L.) and their Expression Profiles in Response to Abiotic Stresses
Authored by:
Ruining Li*, †, Boling Zhang†, Ting Li†, Xuyang Yao, Tingting Feng, Hao Ai, Xianzhong Huang*
The manuscript requires several changes to be published.
1.- The introduction needs several references to explain the mode of action of BZR gene family coupled with brassinesteroid signaling.
References
Oh, E., Zhu, J. Y., Ryu, H., Hwang, I., & Wang, Z. Y. (2014). TOPLESS mediates brassinosteroid-induced transcriptional repression through interaction with BZR1. Nature communications, 5(1), 4140.
Zhu, W., Jiao, D., Zhang, J., Xue, C., Chen, M., & Yang, Q. (2020). Genome-wide identification and analysis of BES1/BZR1 transcription factor family in potato (Solanum tuberosum. L). Plant Growth Regulation, 92, 375-387.
Zhu, W. J., Chen, F., Li, P. P., Chen, Y. M., Chen, M., & Yang, Q. (2019). Identification and characterization of brassinosteroid biosynthesis and signaling pathway genes in Solanum tuberosum. Russian Journal of Plant Physiology, 66, 628-636.
Huang, S., Zheng, C., Zhao, Y., Li, Q., Liu, J., Deng, R., ... & Wang, X. (2021). RNA interference knockdown of the brassinosteroid receptor BRI1 in potato (Solanum tuberosum L.) reveals novel functions for brassinosteroid signaling in controlling tuberization. Scientia Horticulturae, 290, 110516.
Han, Y., Yang, R., Zhang, X., Wang, Q., Wang, B., Zheng, X., ... & Bi, Y. (2022). Brassinosteroid accelerates wound healing of potato tubers by activation of reactive oxygen metabolism and phenylpropanoid metabolism. Foods, 11(7), 906.
Zhu, W. J., Chen, F., Li, P. P., Chen, Y. M., Chen, M., & Yang, Q. (2019). Identification and characterization of brassinosteroid biosynthesis and signaling pathway genes in Solanum tuberosum. Russian Journal of Plant Physiology, 66, 628-636.
In the Table 1, the Gen ID given to each of the 8 BZR genes has to be changed. And BZR1 gene seems to be wrong and StBZR7 checked.
StBZR1
Beta-amylase 7; Beta-amylase
Identifier: PGSC0003DMT400000485
StBZR2
BES1/BZR1 homolog protein 2-like; BRASSINAZOLE-RESISTANT 1 protein
Identifier: PGSC0003DMT400011470
StBZR3
BES1/BZR1 homolog protein 4; BRASSINAZOLE-RESISTANT 2 protein
Identifier: PGSC0003DMT400073250
StBZR4
BES1/BZR1 homolog protein 4-like; BRASSINAZOLE-RESISTANT 2 protein
Identifier: PGSC0003DMT400071511
StBZR5
Protein brassinazole-resistant 1-like; Mature anther-specific protein LAT61
Identifier: PGSC0003DMT400071585
StBZR6
BES1/BZR1 homolog protein 4
Identifier: PGSC0003DMT400018282
StBZR7
Ethylene-responsive transcription factor erf084-like; Beta-amylase; Belongs to the glycosyl hydrolase 14 family
Identifier: PGSC0003DMT400062050
StBZR8
Protein brassinazole-resistant 1-like; Mature anther-specific protein LAT61
Identifier: PGSC0003DMT400039879
3.- What is the phenotype of BZR gene mutants.
4.- Are sterol metabolism linked with BZR genes?
5.- What metabolic process interacts with BZR genes?
6.- Do the BZR gene family is involved in tuber formation?
Round 2
Reviewer 3 Report
Comments and Suggestions for Authors
Dear authors I have read your comments in the second round. The mansucrispt was improved, and for me is perfect.
Author Response
Comments and Suggestions for Authors: Dear authors I have read your comments in the second round. The mansucrispt was improved, and for me is perfect.
Response: Thank you for your affirmation! We will continue to revise the manuscript better!